# Determination of the Primary Excitation Spectra in XPS and AES

**DOI:** 10.3390/nano13020339

**Published:** 2023-01-13

**Authors:** Nicolas Pauly, Francisco Yubero, Sven Tougaard

**Affiliations:** 1Université Libre de Bruxelles, Service de Métrologie Nucléaire (CP 165/84), 50 av. F. D. Roosevelt, B-1050 Brussels, Belgium; 2Institute of Materials Science of Sevilla (CSIC–Univ. Sevilla), Av. Américo Vespucio 49, E-41092 Sevilla, Spain; 3Department of Physics, Chemistry and Pharmacy, University of Southern Denmark, DK-5230 Odense, Denmark

**Keywords:** XPS, photoelectron spectroscopy, Auger spectroscopy, primary-excitation spectrum, dielectric response, intrinsic/extrinsic electron energy losses

## Abstract

This paper reviews a procedure that allows for extracting primary photoelectron or Auger electron emissions from homogeneous isotropic samples. It is based on a quantitative dielectric description of the energy losses of swift electrons travelling nearby surfaces in presence of stationary positive charges. The theory behind the modeling of the electron energy losses, implemented in a freely available QUEELS-XPS software package, takes into account intrinsic and extrinsic effects affecting the electron transport. The procedure allows for interpretation of shake-up and multiplet structures on a quantitative basis. We outline the basic theory behind it and illustrate its capabilities with several case examples. Thus, we report on the angular dependence of the intrinsic and extrinsic Al 2s photoelectron emission from aluminum, the shake-up structure of the Ag 3d, Cu 2p, and Ce 3d photoelectron emission from silver, CuO and CeO_2_, respectively, and the quantification of the two-hole final states contributing to the L_3_M_45_M_45_ Auger electron emission of copper. These examples illustrate the procedure, that can be applied to any homogeneous isotropic material.

## 1. Introduction

X-ray photoelectron spectroscopy (XPS) is currently among the most heavily used analytical techniques to obtain the elemental and chemical composition as well as the electronic structure of atoms at the upmost nanoscale surface region of materials [1,2]. This technique became very popular in the eighties when first electron spectrometers got commercially available. Typically, XPS analysis is performed in ultra-high vacuum conditions using as excitation source X-rays, from either Al or Mg anodes in conventional surface analysis labs or monochromatized tunable photon energy below ~1500 eV at synchrotron radiation facilities.

In the last 10–15 years, a big technological effort has been made to perform this type of surface analysis at near ambient pressures [3] (very interesting for catalysis or environmental based studies) or to study buried material layers (using hard X-rays from either synchrotron radiation sources or newly available Cr anodes) [4].

XPS spectra consist of the energy distribution of emitted photoelectrons after excitation by X-rays and transport out of the material. Thus, energy loss processes experienced by the electrons during their trajectories out of the material towards the detector affect the shape and intensity of the peaks in the finally collected photoelectron spectra.

An intuitive way to describe the physics behind the electron emission after the photon absorption of an atom is provided by a two-step model that splits the process into subsequent events [5]. First, photoexcitation of a core electron with the sudden creation of a static core hole and the photoelectron itself, whose kinetic energy is, in a first approximation, the difference between the photon energy and the binding energy of the core electron. Second, the transport of the photoelectron through the material, crossing the surface/vacuum interface towards the electron analyzer.

Primary excitation spectra in XPS are the result of the first event described above. They include processes such as lifetime broadening, spin-orbit coupling, multiplet splitting or shake-up effects linked to eventual different final states of the photoemission process itself. On top of this, the electron transport induces a background of energy-loss structures with contribution from the so-called “intrinsic” and “extrinsic” excitations [6,7]. The former is due to the interaction of the photoelectron with the electric field set up by the core hole in the medium. The latter is due to the interaction of the photoelectron with the electric field set-up by itself in the medium.

A summary of several models that have been applied to describe these effects can be found in ref. [7]. It should be noted that intrinsic and extrinsic excitations are inherent parts of the photoexcitation process and cannot be separated experimentally. Therefore, strictly speaking, experimental photoelectron spectra after removal of the background due to only extrinsic energy losses, should not be quantitatively compared to these theoretical evaluations [8].

In this line, it is important to have procedures based on the physics behind electron transport in matter that remove both extrinsic and intrinsic inelastic background excitations from experimental data. They will contribute to enhance the quality and the understanding of first principle calculations of spin-orbit, multiplet splittings and shake up effects.

In the past, the inelastic background in XPS has been used extensively to obtain quantitative elemental information on nanostructured surfaces [1]. In the present paper, we intend to remove the background to allow for a better understanding of the basic excitation mechanisms linked to the photoemission process.

In 1998, a dielectric response model was proposed to describe both intrinsic and extrinsic excitations of swift electrons travelling nearby surfaces in presence of stationary positive charge [9]. In this model, the response of a semi-infinite homogeneous isotropic medium is described by a non-local complex dielectric function. The photoelectron and the core hole are suddenly created at time zero at the same location inside the medium. In these conditions, the induced potential created by the static hole and the moving electron is calculated solving the corresponding Poisson’s equation in Fourier space. This allows for evaluating an average effective energy-differential cross section for inelastic electron scattering specific for the process that allows for a straightforward analysis of experimental spectra to obtain the corresponding primary excitation signal.

A clear advantage of using a dielectric-response description to obtain the primary excitation signal from experimental spectra is that it can readily be applied to any material just by considering appropriate complex dielectric function of the medium where the sudden electron-hole creation and photoelectron transport takes place. In addition, it is expected that this approach supplies more reliable results than phenomenological inelastic background subtraction approaches to interpret photoelectron or Auger electron emissions in a quantitative basis.

The purpose of this paper is to briefly describe the dielectric response model [9] to evaluate the corresponding inelastic scattering cross sections as well as the procedure to obtain either XPS or AES primary excitation spectra from experimental data. We also illustrate the capabilities of this procedure with the analysis of several characteristic photoelectron emissions and include quantitative comparison to first principle quantum mechanical calculations.

Finally, we want to emphasize that the theory behind the dielectric response models outlined in this review paper has been implemented in freely available software package QUEELS-XPS [10], to let other interested users apply the theory to the study of photoelectron or Auger emissions of other materials. In the particular case of the analysis of primary emission spectra, it is very much extended within the scientific community the use of locally developed phenomenological methods to prepare experimental data (mostly for background subtraction) for their comparison with theoretical evaluations. Although these phenomenological methods [11,12] allow for qualitative studies, their validity when claiming quantitative comparison between theory and experiment is questionable. Thus, we hope that QUEELS-XPS help to improve quantitative analysis of primary emission spectra because it is based on electron transport theory and specific electron energy material response.

## 2. Theoretical Model

The procedure considered here to extract the energy distribution of the primary photoelectron emission spectra from experimental data is based on two main assumptions:(a)A non-local dielectric description of the material response [9]. This allows for a straightforward analytical evaluation of the energy loss probability of swift emitted electrons in presence of a positive stationary charge;(b)Elastic scattering is neglected. This is justified by the fact that it is of minor importance for the description of energy losses lower than ~30 eV [13], which are mostly considered here.

Under these conditions, the primary emission *F*(*E*) from an isotropic homogeneous sample can be obtained by the expression [14]
(1)$$F\left(E\right)\;≅\;J\left(E\right)-\lambda_{sc}\int_{E}^{\infty }K_{eff,av}^{XPS}\left(E^{\prime }-E\right)J\left(E^{\prime }\right)dE^{\prime }\;$$
where J(E) is the measured spectrum, $K_{eff,av}^{XPS}\left(E^{\prime }-E\right)$ is the average effective inelastic scattering cross section corresponding to the emitted photoelectrons (see below), *E’ − E* is the energy loss *ħ*ω, and *λ_sc_* is the electron inelastic mean free path defined as the inverse of the area of $K_{eff,av}^{XPS}\left(\hbar \omega \right)$. If $K_{eff,av}^{XPS}\left(\hbar \omega \right)$ is known, *F*(*E*) can be evaluated by Equation (1), which is also one of the cases implemented in the QUASES Quantitative Analysis of Surfaces by Electron Spectroscopy software package [15].

$K_{eff,av}^{XPS}\left(\hbar \omega \right)$ is evaluated by modeling the energy losses of the emitted photoelectron (emission in presence of a static positive charge nearby a material–vacuum interface) within a dielectric description of the medium where the electron transport takes place. This modeling is briefly summarized below.

Let us consider a sudden creation of an electron-hole pair at a depth *x* below the surface of a semi-infinite medium. Let us assume that the emitted photoelectron travels along a straight line with velocity *v*, energy *E*_0_ and emission angle *θ*, while the core hole is stationary (infinite lifetime). The effective inelastic electron scattering cross section $K_{eff}^{XPS}\left(x;\hbar \omega \right)$ defined as the probability that the electron, excited at depth *x*, loses an energy *ħω* per unit energy loss and per unit path length traveled inside the solid can be evaluated from the induced potential $\Phi_{ind}\left(x;k,\omega \right)$ created by the static hole and the moving electron according to the expression [9]:(2)$$K_{eff}^{XPS}\left(x;\hbar \omega \right)=\frac{-1}{8\pi^{4}x\hbar^{2}\omega }\int dt\int dr\rho_{e}\left(r,t\right)\;Re\left\{i\int dk\;kv\;\Phi_{ind}\left(x;k,\omega \right)e^{i\left(kr-\omega t\right)}\right\}$$
where ***k*** is the transferred momentum, ***r*** the electron position, *ρ_e_*(***r***,*t*) the charge density of the electron, *t* the time (at *t* = 0 the electron-hole pair is created), and *Re*{} refers to the real part of the quantity in brackets. $\Phi_{ind}\left(x;k,\omega \right)$ is obtained within the surface reflection model [9]. The detailed final analytical expression for $K_{eff}^{XPS}\left(x;\;\hbar \omega \right)$ is rather involved (c.f. ref. [9]). At this stage, it is worth emphasizing that the material response is included in Equation (2) through its complex dielectric function required to evaluate $\Phi_{ind}\left(x;k,\omega \right)$. Thus, the electron energy losses caused by the sudden electron-hole pair creation and subsequent photoelectron transport out of the material gives rise to both collective excitations (i.e., plasmons) and interband electron transitions.

The algebra required to obtain an analytical expression for $K_{eff}^{XPS}\left(x;\hbar \omega \right)$ from Equation (2) allows for separating the electron energy losses linked to the interaction of the moving electron with the electric field set up by the static positive charge (i.e., intrinsic energy losses) from those linked with the electric field set-up by the moving electrons itself. (i.e., extrinsic energy losses).

The dielectric response of the medium is described by the so-called energy loss function (ELF), i.e., the imaginary part of the inverse of the dielectric function *Im*{−1*/ϵ*(*k*, *ħω*)}. It is usually parametrized as addition of oscillators [9]
(3a)$$Im\left\{\frac{-1}{\varepsilon \left(k,\hbar \omega \right)}\right\}=\theta \left(\hbar \omega -E_{g}\;\right)\cdot \overset{n}{\underset{i=1}{\sum }}\frac{A_{i}\;\hbar \gamma_{i}\;\hbar \omega }{{\left[{\left(\hbar \omega_{0ik}\right)}^{2}-{\left(\hbar \omega \right)}^{2}\right]}^{2}+{\left(\hbar \gamma_{i}\right)}^{2}\;{\left(\hbar \omega \right)}^{2}}\;$$
with the momentum dispersion relation
(3b)$$\hbar \omega_{0ik}=\hbar \omega_{0i}+\alpha_{i}\frac{\hbar^{2}k^{2}}{2m}$$
where *A_i_*, *ħω_0i_,*
*ħγ_i_ and α_i_* denote the strength, position, width and dispersion parameters of the *i*’th ELF oscillator. The step function *θ*(*ħω* − *E_g_*) (*θ* = 1 for $\hbar \omega >E_{g}$; *θ* = 0 for $\hbar \omega <E_{g}$) is included to describe the band gap energy *E_g_* in semiconductors and insulators. The parameters describing the energy loss function may be taken from the literature [16,17], or they can conveniently be determined from analysis of reflection electron energy loss spectroscopy (REELS) measurements [18].

In XPS measurements from homogeneous samples, electrons from a wide range of depths are detected. Thus, the specific inelastic scattering cross-section $K_{eff,av}^{XPS}\left(\hbar \omega \right)$ will be a weighted average of $K_{eff}^{XPS}\left(x;\hbar \omega \right)$ over path-lengths *x*, according to,
(4)$$K_{eff,av}^{XPS}\left(\hbar \omega \right)=\int_{0}^{\infty }dxW\left(x\right)K_{eff}^{XPS}\left(x;\hbar \omega \right)$$
where *W*(*x*) is the path length distribution for those electrons that have undergone a single inelastic collision. At this point, it is worth to recall that $K_{eff,av}^{XPS}\left(\hbar \omega \right)$ includes losses during transport in the bulk, through the surface region and core hole induced losses as well as interferences between these effects.

For practical purposes, $K_{eff,av}^{XPS}\left(\hbar \omega \right)$ can be evaluated using the freely available QUEELS-XPS software (QUantitative analysis of Electron Energy Losses at Surfaces for XPS) [10,19]). The input parameters of the calculation are the initial kinetic energy *E*_0_ and the angle of emission *θ* of the photoelectron, and the parametrized energy-loss function, which characterizes the dielectric response of the material.

Figure 1 shows, as an example, a series of $K_{eff}^{XPS}\left(x;\hbar \omega \right)$ and the corresponding $K_{eff,av}^{XPS}\left(\hbar \omega \right)$ calculated for 1000 eV photoelectrons emitted in presence of a stationary charge at normal emission in silicon. $K_{eff,av}^{XPS}\left(\hbar \omega \right)$ is large for energy losses smaller than approximately 5 eV and diverges for zero energy loss. This is mostly due to the excitation of electrons in the conduction band and leads to the asymmetric line shape in the XPS spectra for metals (the so-called Doniach-Sunjic line shape [20]).

As a final step, after the $K_{eff,av}^{XPS}\left(\hbar \omega \right)$ has been evaluated, the primary emitted spectrum *F*(*E*) obtained from Equation (1) may be decomposed as the addition of several *F_i_*(*E*) peaks that account for the local quantum effects not included in the dielectric description of the material, such as spin-orbit, multiplet splitting and shake-ups.

An alternative way to obtain the primary emitted spectrum *F*(*E*) consist in fitting experimental *J*(*E*) data by a trial-and-error procedure through the following series [21]
(5)$$J\left(E\right)\;≅\;\int_{0}^{\infty }dE_{0}F\left(E_{0}-E\right)\left[\delta \left(E_{0}\right)+\lambda K_{eff,av}^{XPS}+\frac{\lambda^{2}}{2}K_{eff,av}^{XPS}⊗\;K_{eff,av}^{XPS}+\ldots \right]$$
where ⊗ is the sign for convolution and *F*(*E*) is a parametrized function.

Note that the methods briefly described above can also be used to calculate X-ray excited AES (X-ray excited Auger electron spectroscopy) primary excitation spectra from experimental data. Indeed, the effective inelastic scattering cross section corresponding to Auger electron emission in presence of two static holes can be simulated by considering that the electron emission takes place in the presence of a static positive charge with twice the charge of a single hole.

## 3. Case Studies

In the following, we present five case examples where we have applied the dielectric response model briefly described above to quantify the energy loss processes related to the electron transport nearby a surface in presence of stationary positive charge. First, we report on the angular dependence of the Al 2s photoelectron emission from metallic aluminum [22]. The next three examples concern the complex shake-up structure of the Ag 3d, Cu 2p, and Ce 3d photoelectron emission from Ag [23], CuO [24] and CeO_2_ [25], respectively. The last example reports the quantification of the two-hole multiplet final states contributing to the L_3_M_45_M_45_ Auger emission from metallic copper [26]. In these case examples, the analysis of electron emitted spectra has been performed according to either Equation (1) or Equation (5) depending on the convenience to illustrate particular effects of the material under study.

### 3.1. Angular Dependence of Al 2s Photoelectron Emission from Aluminum

Photoelectron emission from aluminum has been widely reported in the past. This is mainly due to the sharp single peak energy loss function describing the dielectric behavior of this material, which allows for “easy” identification of surface and bulk plasmon excitations. As an example of these studies, ref. [22] reports a comparison between experimental evaluations of Al 2s photoelectrons from aluminum excited with Mg Kα radiation and the corresponding model calculations using the procedure outlined above. The main purpose of this early paper was to test whether the dielectric response description of the electron energy losses emitted in presence of a stationary positive charge was able to reproduce the angular dependence of the partition between surface and bulk plasmon excitations observed experimentally in this material. Figure 2a show these *J*(*E*) model calculations using Equation (5) that have to be compared with the corresponding experimental data shown in Figure 2b (adapted from ref. [27]). As expected, the relative contribution of surface to bulk losses increases for increasing emission angles. Note that the model calculations reproduce the absolute intensities as well as shapes of both bulk and surface excitations for all emission angles considered. This result gives confidence to the validity of the model.

The theoretical evaluation of the cross section outlined above allows to calculate separately the extrinsic and intrinsic contributions to the total cross section. Figure 2c shows $K_{eff,av}^{XPS}\left(\hbar \omega \right)$ evaluated for photoelectrons of 1130 eV travelling in Al at increasing emission angles. Intrinsic and extrinsic contributions to the cross section are defined as loss terms during the photoelectron emission induced by either the presence of the hole or independent of it. Note that surface losses at ~10 eV are mostly due to extrinsic losses while both intrinsic and extrinsic excitations contribute significantly to bulk losses at ~15 eV. Moreover, the shape of the intrinsic (surface and bulk) losses is clearly asymmetric while the extrinsic surface and bulk losses have only a small asymmetry. Note also that the relative contribution of intrinsic excitations decreases with increasing emission angle. This can be explained by considering that, for glancing emission angles, the photoelectron interacts for a longer time with the evanescent field set up at the material surface compared to photoelectrons emitted normally to the surface. More detailed analysis can be found in ref. [22].

### 3.2. Ag 3d Photoelectron Emission from Silver

Silver is another material whose energy loss structure as obtained from photoelectron emission experiments has been extensively studied [28]. It shows several weak loss features on the low kinetic energy side of the Ag 3d photoelectron peaks. Figure 3 shows a comparison between experimental and simulated Ag 3d spectra [23]. It also includes the originally excited spectrum *F*(*E*) considered in the simulation. In this case it was required that the *F(E)* spectrum includes, besides the main Ag 3d_5/2_ and Ag 3d_3/2_ peaks, shake up contributions shifted by 13.2 eV and 18.0 eV from the main 3d peaks that account for approximately 1–3% of their intensity. This result agrees with quantum mechanical calculations that assigned similar contributions to 4d → 5p and 4d → 5s final state shake-ups [29]. Note that the quantification of these features would hardly be possible with straight line or Shirley background analysis typically applied to isolate photoelectron peaks.

### 3.3. Ce 3d Photoelectron Emission from CeO_2_

The analysis of Ce3d from CeO_2_ has also been rather controversial in the past due to the complex photoemission spectra of this material [30,31,32,33]. Figure 4 shows raw experimental XPS data of this emission together with the primary excited spectra obtained after subtraction of the inelastic background determined within either the Shirley method [11], or the present model using the effective inelastic-scattering cross section for XPS, $K_{eff,av}^{XPS}\left(\hbar \omega \right)$, considering both extrinsic and intrinsic material-specific excitations. Note that the choice of the procedure for inelastic background subtraction clearly influences the relative intensity of the different features within the primary spectrum. We emphasize that even though the Shirley modeling of background correction often gives reasonably “good looking” results when used to isolate peaks, it is questionable to apply it for complex photoemitted peak line shapes, such as Ce 3d from CeO_2_ [34].

The Ce *3d* structure of Ce^4+^ oxidation state in CeO_2_ is often subdivided into 6 peaks labelled v, v″, v‴, u, u″ and u‴ [30], v and u referring to the 3*d*_5/2_ and 3*d*_3/2_ spin–orbit components, respectively. The doublet v/u corresponds to the final state Ce3*d*^9^*4f*
^2^O*2p*^4^, while v″/u″ are assigned to Ce3*d*^9^*4f*^1^-O2*p*^5^ and v‴/u‴ to Ce3*d*^9^*4f*^0^-O*2p*^6^, respectively [32]. The interpretation of the v/u components has been rather controversial in the past. Thus, for example Skála et al. [33] proposed either the use of asymmetric tails for v and u, or the modeling of v and u by two symmetric peaks. They found better consistency for the second interpretation that, in fact, is supported by Dirac–Fock and configuration-interaction wave-functions calculations [35]. This last interpretation is consistent with the analysis reported in Figure 4.

### 3.4. Cu 2p Photoelectron Emission from CuO

It is well known that the Cu 2p photoelectron emission from CuO is characterized by a pronounced shake up structure at approximately 7–9 eV higher binding energies than the main Cu 2p lines [36] that is explained within a molecular orbital description by a charge transfer mechanism. On the one hand, the main lines are attributed to a *c*3*d*^10^*L* (*c* denotes a core hole and *L* a hole in the ligand) final state in which an electron is transferred from the ligand into the Cu 3*d* level. On the other hand, the shake up is attributed to a *c*3*d*^9^ final state in which no charge transfer occurs [36].

Figure 5 shows experimental *J*(*E*) spectrum of the Cu 2p emission from CuO and the corresponding model calculation evaluated according to Equation (5), as well as the remarkable quantitative agreement between the *F*(*E*) obtained from the fitting analysis and its evaluation within the charge transfer multiplet model implemented in CTM4XAS software [8] using the input parameters reported in ref. [37]. Note that the only input for the evaluation of the *F*(*E*) from the experimental data is the corresponding CuO energy loss function.

### 3.5. L_3_M_45_M_45_ Auger Emission from Copper

The method outlined in Section 2 can also be used to quantify primary emitted Auger spectra excited by energetic electrons or X-rays. This was completed in ref. [26] as an alternative to using empirical fitting spectral deconvolution procedures [12].

Figure 6 shows the experimentally detected L_3_M_45_M_45_ Auger emission from copper excited with Mg Kα X-rays. It mainly arises from the *L*_3_ (2*p*_3*/*2_) core-hole decay via the Auger process involving two *M*_45_ (3*d*) electrons resulting in a final 3*d*^8^ configuration. We then may expect to observe, due to L-S coupling, a multiplet emission with five terms namely ^3^F, ^1^D, ^3^P, ^1^G and ^1^S. This contribution is referred to as the “normal” Auger contribution to the measured *L*_3_*M*_45_*M*_45_ transition. It comes out that the observed spectrum has extra features compared with the theoretically evaluated “normal” Auger emission [38]. In fact, there is some contribution in the low kinetic energy part of the *L*_3_*M*_45_*M*_45_ spectrum [38,39,40,41] due to a *L*_2_*L*_3_*M*_45_ Coster–Kronig transition. Indeed, after a *L*_2_*L*_3_*M*_45_ Coster–Kronig transition, a *L*_3_*M*_45_*M*_45_ process takes place with a third hole in the *M*_45_ level of the final state. This extra *L*_3_*M*_45_*M*_45_ process is thus shifted to lower kinetic energy because of the Coulomb interaction between this *M*_45_ spectator vacancy and the Auger electron. The multiplet structure associated with such an Auger vacancy satellite transition corresponds to a 3*d*^7^ configuration. We will refer to this contribution as the “extra” Auger contribution to the measured *L*_3_*M*_45_*M*_45_ transition.

Figure 6 shows experimental *J*(*E*) spectra and the corresponding simulation obtained with the method described in Section 2 as well as the decomposition of the different terms contributing to the primary Auger emission. Each peak is labeled in accordance with the Russell-Saunders terminology. The * symbol identify the Auger vacancy satellite structure. Below the theoretical intensities of the “normal” individual term calculated from perturbation theory [12,38] are shown for comparison purposes. Thus, a ~5.1 eV energy shift is observed between the main “normal” and the “extra” Auger emission. On the other hand, the intensity ratio of the “extra” to the “normal” Auger emissions is 40%. These results compare well with those reported by Antonides et al. [38].

## 4. Summary and Concluding Remarks

This paper reviews a procedure that allows for extracting primary photoelectron or Auger electron emissions from homogeneous isotropic samples. It is based on a quantitative dielectric response description of the energy losses of swift electrons travelling nearby surfaces in presence of static positive charges. The theory behind the modeling of the electron energy losses takes into account intrinsic and extrinsic effects affecting the electron transport. The model has been implemented in the freely available QUEELS-XPS software package [10], which requires as only input the energy loss function of the material that may be taken from the literature or may easily be determined experimentally by analysis of a REELS spectrum from the material.

The procedure outlined here can be considered as a quantitative alternative to other phenomenological procedures often used to remove inelastic background from XPS or AES experimental data to obtain the true primary excitation spectra. It allows for the interpretation of shake-up and multiplet structures on a quantitative basis. The examples included in this review have been selected because they mostly correspond to well-understood systems and therefore serve as a good test of the validity of the dielectric response approach. The method applies to any homogeneous isotropic material, including melts, glasses or polymers provided their energy loss function is known. If this function is not available, it can be determined from analysis of a REELS spectrum of the studied material.

## Figures and Tables

**Figure 1 nanomaterials-13-00339-f001:**
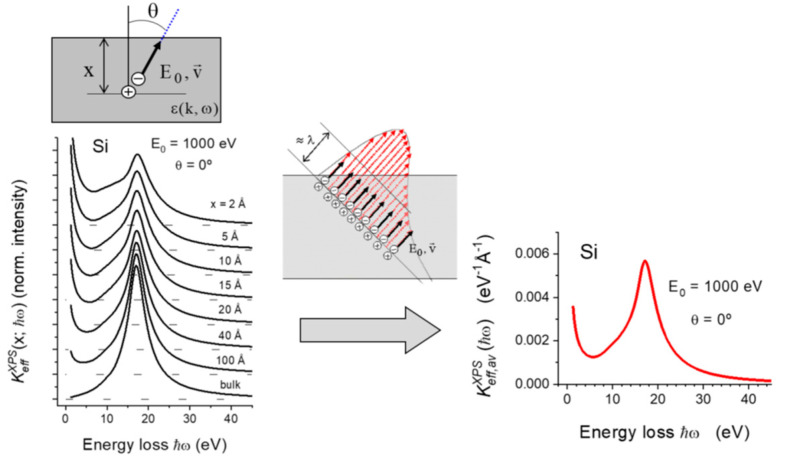
Schematic representation of the process to obtain average effective inelastic scattering cross sections within the dielectric model outline above. (**Left**): $K_{eff}^{XPS}\left(x;\hbar \omega \right)$ for 1000 eV photoelectrons excited at several depths *x* in Silicon. (**Right**): The corresponding $K_{eff,av}^{XPS}\left(\hbar \omega \right)$ evaluated according to Equation (4) as illustrated in the upper right part.

**Figure 2 nanomaterials-13-00339-f002:**
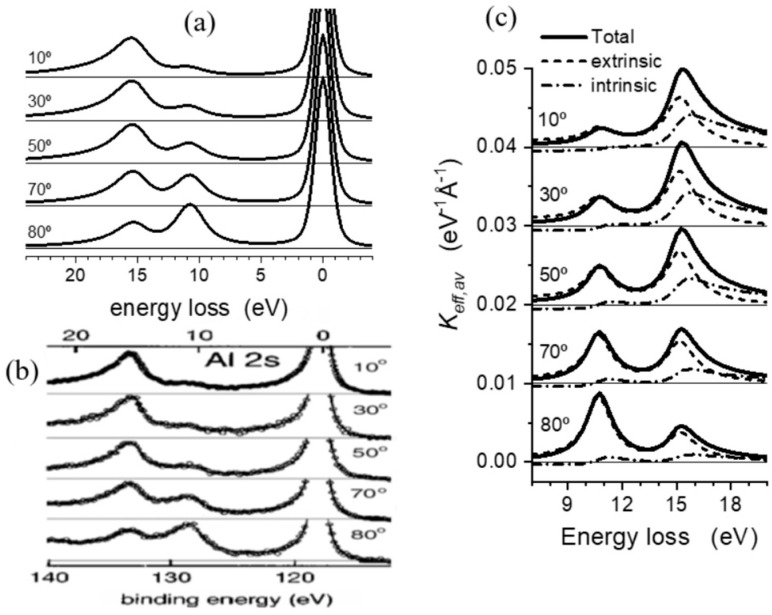
(**a**) Theoretical and (**b**) experimental (adapted from Biswas et al. [27]) evaluation of Al 2s photoelectron emission from aluminum for several emission angles. (**c**) Corresponding $K_{eff,av}^{XPS}\left(\hbar \omega \right)$ cross sections together with their decomposition into extrinsic (dashed lines) and intrinsic (dashed-dotted lines) contributions.

**Figure 3 nanomaterials-13-00339-f003:**
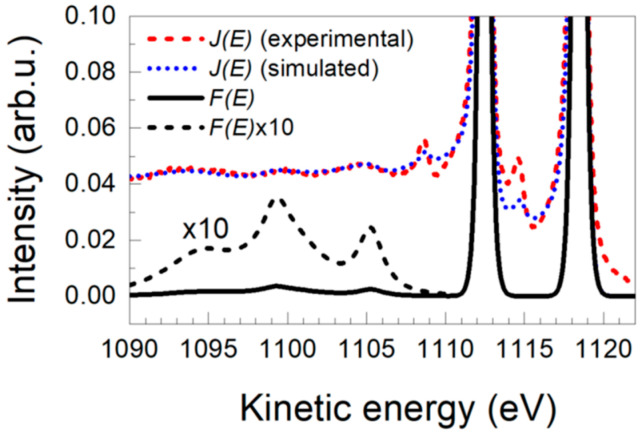
Blow up of the shake-up structure corresponding to the Ag 3d photoelectron emission of silver. The figure includes experimental data (red dashed line), model calculation (blue dashed-dotted line) and the corresponding initially excited *F*(*E*) spectrum considered in the calculation (full black line). The spectra are normalized so the Ag 3d_5/2_ peak has unit area.

**Figure 4 nanomaterials-13-00339-f004:**
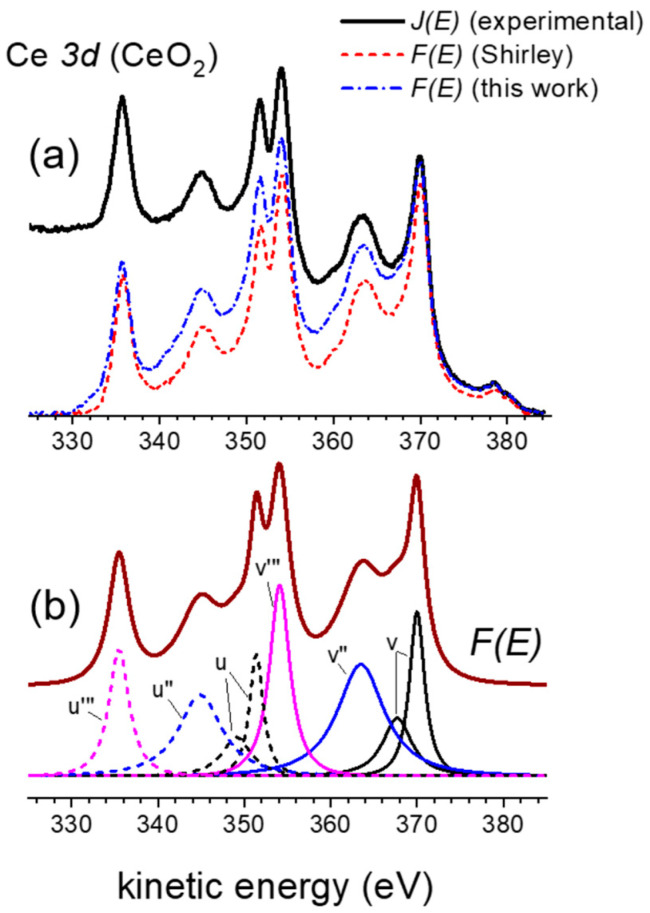
(**a**) Experimental *J*(*E*) spectra (thick solid line), and corresponding primary excitation spectra *F*(*E*) evaluated by means of Shirley background correction or using Equation (1). (**b**) Decomposition of *F*(*E*) in its Ce 3d_5/2_ (v, v″ and v‴) and Ce 3d_3/2_ (u, u″, and u‴) components.

**Figure 5 nanomaterials-13-00339-f005:**
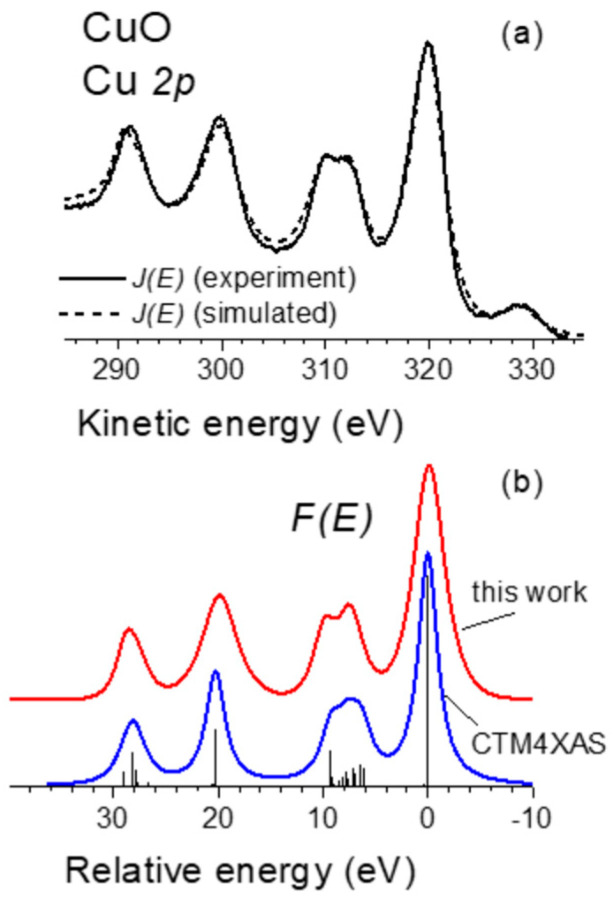
(**a**) Experimental and simulated *J*(*E*) spectra of the Cu 2p emission from CuO. (**b**) Primary excited spectrum *F*(*E*) used to evaluate *J*(*E*) compared with the corresponding charge transfer multiplet calculation with CTM4XAS software [8].

**Figure 6 nanomaterials-13-00339-f006:**
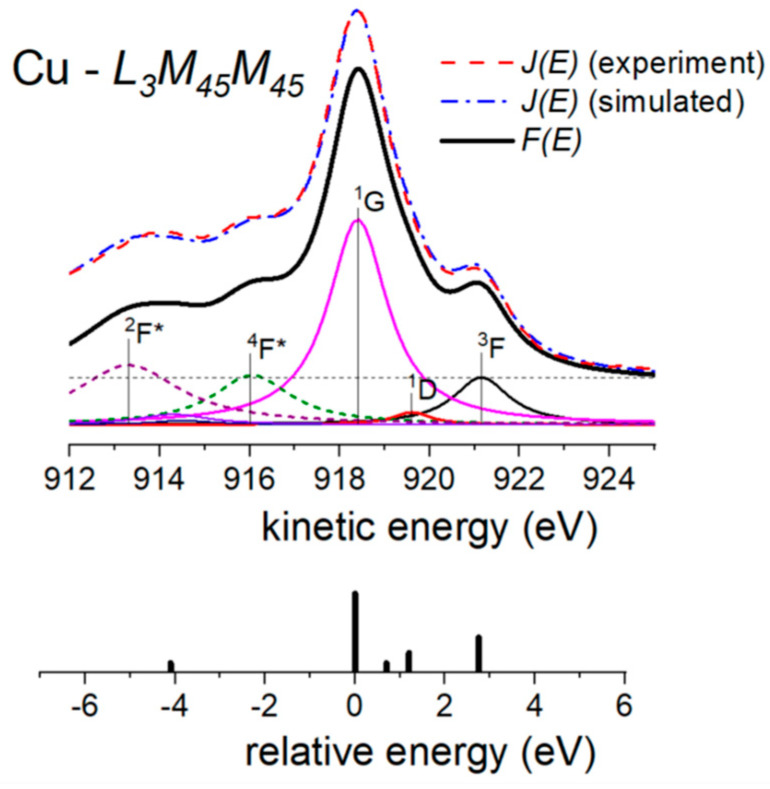
Experimental and simulated *J*(*E*) spectra of the L_3_M_45_M_45_ Auger emission from copper. In addition, the corresponding primary excited spectrum *F*(*E*) is included together with their most intense final state terms (full lines: 3d^8^ normal Auger emission; dashed lines: 3d^7^ extra Auger vacancy satellite structure). Bottom: relative energy and intensity of the 3d^8^ individual terms contributing to the Auger emission.

## Data Availability

All data available on request.

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
