# Peer review of "Determination of the Primary Excitation Spectra in XPS and AES"

_nanomaterials, 2023, doi:10.3390/nano13020339_

Round 1

Reviewer 1 Report

Report on manuscript nanomaterials 2115195

Determination of the primary excitation spectra in XPS and AES
by Nicolas Pauly, Francisco Yubero, and Sven Tougaard

The authors review the QUEELS-XPS software package. They describe the theory behind the calculation of energy losses and Auger electrons as are observed in X-ray photoelectron spectroscopy. The methods are explained using various examples.
The manuscript is well written and good understandable. I suggest publication in its present form after clearing up the error messages (Error! Reference source not found.) in the manuscript.

Author Response

We thank the reviewer for his/her positive report regarding our manuscript. We have thoroughly revised the previous version of the manuscript searching for typos and error messages.

Reviewer 2 Report

The authors are well known by the scientific community because they published in the past years many papers about the topics of the present work. Indeed, this review article contains materials from previous published papers and consequently self-citation is often present.

This work, intended as a review article, surely deserves publication because it summarizes important results on the interpretation of XPS and Auger spectra.

There are the following typos/remarks for the text

Row    56                                          The latter is due…

            121                                        J(E) is not defined in the text

            115-119-128-140-160-185     Error! Reference source not found – this should be corrected

            172-193                                  calls to Ref 13-14-15-19 are missing in the text

            291                                         In Fig.4 letters a) and b) for the caption are missing

            323                                         Copper not “Cupper”

Author Response

We thank the reviewer for his/her positive report regarding our manuscript. We have thoroughly revised the previous version of the manuscript searching for typos, error messages related to references not found, and missing letters in Fig.4. Besides we have remade the last paragraph in the Summary and Concluding Remarks section as follows

" The procedure outlined here can be considered as a quantitative alternative to other phenomenological procedures often used to remove inelastic background from XPS or AES experimental data to obtain the true primary excitation spectra. It allows the interpretation of shake-up and multiplet structures on a quantitative basis. The examples included in this review have been selected because they mostly correspond to well-understood systems and therefore serve as a good test of the validity of the dielectric response approach. The method applies to any homogeneous isotropic material, including melts, glasses or polymers provided their energy loss function is known. If this function is not available, it can be determined from analysis of a REELS spectrum of the studied material.

We have also added the following sentence at the end of the Abstract:

“These examples illustrate the procedure, that can be applied to any homogeneous isotropic material”

Reviewer 3 Report

This is a nice review with examples for determining primary excitation spectra in XPS and AES data.

What I find missing is a discussion of the scope of materials the dielectric energy loss approach can be applied to. This will be very important for potential users of the technique. The examples chosen are common metal and oxide model systems, which are well understood already and often chosen for demonstration studies, but will not be of much use to many users. Could the authors provide guidance for potential new users about what to do for other systems, or whether the approach is even likely to work for other classes of materials? I am particularly thinking of organic crystaline and polymeric materials, and non-crystalline materials such as melts and glasses. There is in my experience confusion in the community about these issues and it is an opportunity to clarify the scope through this OA review.

The manuscript needs another round of spell checking. Here are some examples I picked up:

Line 46 – in a first approximation (not approach)

Line 115 - Error! Reference source not found

Line 119 - Error! Reference source not found

Line 128 - Error! Reference source not found

Line 140 - Error! Reference source not found

Line 160 - Error! Reference source not found

Line 323 – Cupper should be Copper

Line 344 – Russell-Saunders (not Russel-Sunders)

Author Response

We thank the reviewer for his/her positive report. We have thoroughly revised the previous version of the manuscript searching for typos and error messages. Besides, we have remade the last paragraph in the Summary and Concluding Remarks section to provide guidance for other potential new users of the technique. The new paragraph read as follows

The procedure outlined here can be considered as a quantitative alternative to other phenomenological procedures often used to remove inelastic background from XPS or AES experimental data to obtain the true primary excitation spectra. It allows the interpretation of shake-up and multiplet structures on a quantitative basis. The examples included in this review have been selected because they mostly correspond to well-understood systems and therefore serve as a good test of the validity of the dielectric response approach. The method applies to any homogeneous isotropic material, including melts, glasses or polymers provided their energy loss function is known. If this function is not available, it can be determined from analysis of a REELS spectrum of the studied material

We have also added the following sentence at the end of the Abstract:

 “These examples illustrate the procedure, that can be applied to any homogeneous isotropic material”